# The E2F Pathway Score as a Predictive Biomarker of Response to Neoadjuvant Therapy in ER+/HER2− Breast Cancer

**DOI:** 10.3390/cells9071643

**Published:** 2020-07-08

**Authors:** Masanori Oshi, Hideo Takahashi, Yoshihisa Tokumaru, Li Yan, Omar M. Rashid, Masayuki Nagahashi, Ryusei Matsuyama, Itaru Endo, Kazuaki Takabe

**Affiliations:** 1Department of Surgical Oncology, Roswell Park Comprehensive Cancer Center, Buffalo, NY 14263, USA; masa1101oshi@gmail.com (M.O.); hideo.takahashi@roswellpark.org (H.T.); Yoshihisa.Tokumaru@roswellpark.org (Y.T.); 2Department of Gastroenterological Surgery, Yokohama City University School of Medicine, Yokohama 2360004, Japan; ryusei@yokohama-cu.ac.jp (R.M.); endoit@med.yokohama-cu.ac.jp (I.E.); 3Department of Surgical Oncology, Graduate School of Medicine, Gifu University, Gifu 501-1194, Japan; 4Department of Biostatistics & Bioinformatics, Roswell Park Comprehensive Cancer Center, Buffalo, NY 14263, USA; li.yan@roswellpark.org; 5Department of Surgery, Holy Cross Hospital, Michael and Dianne Bienes Comprehensive Cancer Center, Fort Lauderdale, FL 33308, USA; omarmrashidmdjd@gmail.com; 6Department of Surgery, Massachusetts General Hospital, Boston, MA 02114, USA; 7Division of Digestive and General Surgery, Niigata University Graduate School of Medical and Dental Sciences, Niigata 9518520, Japan; masanagahashi@gmail.com; 8Department of Gastrointestinal Tract Surgery, Fukushima Medical University School of Medicine, Fukushima 9601295, Japan; 9Department of Surgery, Jacobs School of Medicine and Biomedical Sciences, State University of New York, Buffalo, NY 14263, USA; 10Department of Surgery, Niigata University Graduate School of Medical and Dental Sciences, Niigata 9518510, Japan; 11Department of Breast Surgery and Oncology, Tokyo Medical University, Tokyo 1608402, Japan

**Keywords:** biomarker, breast cancer, cell cycle, cyclin-dependent kinase, E2F, gene set, GSVA, hormone receptor, immune checkpoint, metastasis, signaling pathway, treatment response

## Abstract

E2F transcription factors play critical roles in the cell cycle. Therefore, their activity is expected to reflect tumor aggressiveness and responsiveness to therapy. We scored 3905 tumors of nine breast cancer cohorts for this activity based on their gene expression for the Hallmark E2F targets gene set. As expected, tumors with a high score had an increased expression of cell proliferation-related genes. A high score was significantly associated with shorter patient survival, greater MKI67 expression, histological grade, stage, and genomic aberrations. Furthermore, metastatic tumors had higher E2F scores than the primary tumors from which they arose. Although tumors with a high score had greater infiltration by both pro- and anti-cancerous immune cells, they had an increased expression of immune checkpoint genes. Estrogen receptor (ER)-positive/human epidermal growth factor receptor 2 (HER2)-negative cancer with a high E2F score achieved a significantly higher pathological complete response (pCR) rate to neoadjuvant chemotherapy. The E2F score was significantly associated with the expression of cyclin-dependent kinase (CDK)-related genes and strongly correlated with sensitivity to CDK inhibition in cell lines. In conclusion, the E2F score is a marker of breast cancer aggressiveness and predicts the responsiveness of ER-positive/HER2-negative patients to neoadjuvant chemotherapy and possibly to CDK and immune checkpoint inhibitors.

## 1. Introduction

Cell cycle progression is a critical component of cell proliferation, and continuous proliferation is one of the hallmarks of cancer [1]. Components regulating the E2F pathway have been identified in nearly every human malignancy, and many of them including E2F transcription factors themselves play major roles in cancer progression, metastasis, and treatment response of breast cancer [2]. Members of the E2F family of transcription factors are known for controlling the cell cycle. The transcriptional activity of E2Fs is regulated through cyclin-dependent kinase (CDK) 4/6-cyclin D complexes that function as a cell cycle checkpoint, phosphorylating the retinoblastoma (Rb) tumor suppressor protein and thereby releasing E2F1 from Rb to cause the transcription of G1-S transition and S phase genes. These E2F target genes encode proteins involved in DNA replication, cyclins, and the E2F transcription factors themselves. The dysregulated expression of E2Fs is often observed in various malignances, including breast cancer.

Over 70% of breast cancer patients have estrogen receptor (ER)-positive tumors [3], which are generally less aggressive compared to other tumor subtypes. Yet, some challenges remain in this subtype, such as poor response to neoadjuvant chemotherapy (NAC) [4,5,6] and late recurrence that occurs in 40% of patients more than 10 years after diagnosis [7]. With systemic chemotherapy becoming a major component of advanced breast cancer treatment [8], poor response to NAC is especially challenging in this subtype. Although many research studies and clinical trials are ongoing to improve patient outcomes such as the combination of endocrine therapy with different targeted agents that interfere with other signaling pathways and cell-cycle progression [9], no biomarkers have been established to identify those who benefit from NAC in this subtype. Thus, the identification of new biomarkers for better patient selection is of paramount importance. In recent years, drugs that inhibit CDK4/6 activity (palbociclib, ribociclib, or abemaciclib [10,11,12,13]) in combination with endocrine therapy have been used to treat patients with ER-positive/human epidermal growth factor receptor 2 (HER2)-negative advanced breast cancer. Results from clinical trials suggest that the E2F transcriptional pathway is an important factor for treatment with CDK4/6 inhibitors in breast cancer. However, with myriads of other genes involved in the E2F pathway, it is difficult to comprehend the pathway from a single-gene perspective.

Rapid evolution in microarray and sequencing technologies has revolutionized the depth and complexity at which molecular data is obtained and examined today in biomedical research. For example, computational models utilizing the expression levels of multiple genes [14] can predict cancer drug sensitivity even if at the single gene level, such prediction has been difficult to reproduce across independent studies and interpret biological meaning [15]. Since there is growing evidence that multiple genes are involved in drug response [16], competitive scoring capturing multiple pathways and multiple genes can provide more accurate predictions than self-contained scoring [17]. With a pathway or gene set-based approach, such coordination of genes can be taken into account, reducing model complexity and increasing the explanatory power of prediction models [18,19,20]. Gene Set Variant Analysis (GSVA) is an analysis method that explores the biological activity of a signaling pathway of interest, and this method has been widely used to score pathway activity from global gene expression data [17,21,22,23,24]. 

Given this feature of GSVA, we used it in this study to dichotomize patients with breast cancer based on the activity of the E2F signaling pathway within their tumors. Recently, we reported that the level of G2M cell cycle pathway activity may serve as a useful tool for identifying patients who are likely to metastasize and have a poor survival in estrogen receptor (ER)-positive breast cancer using GSVA [21]. Given that the G2M checkpoint occurs between the G2 and M phases whereas E2F plays a major role during the G1/S transition, we hypothesized that tumors with increased E2F transcriptional activity, measured by scoring the gene expression of E2F target genes, will have characteristics of more aggressive disease as well as a better response to systemic chemotherapy among patients with ER-positive/HER2-negative breast cancer. 

## 2. Materials and Methods

### 2.1. TCGA and METABRIC Breast Cancer Cohorts and Their Data

The Pan-Cancer Clinical Data Resource [25] and cBio Cancer Genomic Portal [26] provided The Cancer Genome Atlas (TCGA)–breast invasive carcinoma (BRCA) tumoral genomic profiling and clinical information. Data of 1065 patients who were female and had pathological diagnosis of breast cancer were utilized in the study. Since TCGA database does not include Nottingham histological grade, we previously extracted the grades for 573 of the 1065 patients through the manual examination of pathology reports [27]. The Nottingham histological grading score is based on three patient variables: degree of tubular formation, nuclear pleomorphism, and mitosis [28]. The cBioPortal provided tumoral genomic and clinical information from 1903 cases in the Molecular Taxonomy of Breast Cancer International Consortium (METABRIC) cohort [29,30,31]. The prognostic 4-gene score values of the TCGA and METABRIC tumors were previously calculated by us from tumor gene expression of genes *DOK4*, *HCCS*, *PGF*, and *SHCBP1* [32].

### 2.2. Other Breast Cancer Cohorts and Their Data

The Gene Expression Omnibus (GEO) repository of the US National Institutes of Health provided normalized tumoral genomic and clinical data. On an as-needed basis for genes with multiple probes, the mean was used. Log_2_-transformed gene expression data were used for all analyses. The published data of Shi et al. (GSE20194; *n* = 197; regimens: paclitaxel, 5-fluorouracil, cyclophosphamide and doxorubicin) [33], Symmans et al. (GSE25066; *n* = 467; regimens: taxane and anthracycline) [34], Vera-Ramirez et al. (GSE28844; *n* = 33; regimens: anthracycline and taxane) [35], Noguchi et al. (GSE32646; *n* = 81; regimens: 5-fluorouracil/epirubicin/cyclophosphamide and paclitaxel) [36], Massarweh et al. (GSE33658; *n* = 22; regimens: anastrozole, fulvestrant, and gefitinib) [37], and Loi et a. (GSE9195; *n* = 77; regimen: tamoxifen) [38] were obtained to test the association of the E2F score with treatment response. GSE28844 cohort was used, since it is an NAC cohort that transcriptomic data as well as treatment response data (bad, mid, good) are also available before and during NAC. We also used the data of Siegel et al. (GSE110590; *n* = 16 primary tumors and their 46 tumor metastases) [39] and Sinn et al. (GSE124647; *n* = 140 tumor metastases) [40] to investigate the E2F scores in metastatic tumors.

### 2.3. Gene set Expression Analyses

The Gene Set Variation Analysis (GSVA) method [41] was utilized to measure the E2F pathway score as the GSVA score for the E2F targets gene set of the MSigDb Hallmark collection [42] using the GSVA Bioconductor package (version 3.10). High and low E2F score designations were assigned by within-cohort median values. GSEA software (Lava version 4.0) and MSigDb Hallmark gene sets provided gene set enrichment analysis (GSEA) [43,44,45,46,47,48,49,50,51,52,53,54], with a 0.25 false discovery rate (FDR) determining significance. 

### 2.4. Other

Drug sensitivity data for cell lines were obtained from GEO for the GSE36139 study [55]. R software (version 3.6.2) and Excel (version 16 for Windows; Microsoft, Redmond, WA, USA) were the statistics and plot tools used for data analysis. The xCell algorithm provided tumoral compositional analysis for infiltrating immune cells based on tumoral genomic data [56]. A value of 0.05 was the *P* value cut-off for statistical significance. ANOVA or Fisher’s exact tests provided statistical comparisons between groups. In data illustrations, Tukey-type boxplots demonstrate median and inter-quartile level values.

## 3. Results

### 3.1. Tumors with a High E2F Pathway Score Have Enriched Expression of Other Cell Cycle-Related Gene Sets and Are More Aggressive

Using The Cancer Genome Atlas (TCGA)–breast invasive carcinoma (BRCA) cohort, we determined the E2F pathway score of primary tumors as the GSVA score for the Molecular Signatures Database (MSigDb) Hallmark E2F targets gene set [42]. With median value as a cut-off, we divided the TCGA tumors into groups of low and high E2F scores. We hypothesized that tumors with a high score would have enriched expression of genes of other cell cycle-related pathways. To test this hypothesis, we examined the tumor expression of the other Hallmark gene sets using the gene set enrichment analysis (GSEA) method. Tumors with a high E2F score had significantly enriched expression of many other cell cycle-related Hallmark gene sets, including the G2M checkpoint, MYC targets v1 and v2, MITOTIC spindle, MTORC1 signaling, UNFOLDED protein response, and DNA repair (Figure 1A; false discovery rate (FDR) < 0.01). These results were validated with the Molecular Taxonomy of Breast Cancer International Consortium (METABRIC) cohort (Figure 1A; FDR < 0.01), which showed that the E2F pathway score reflects underlying cell proliferation, as expected.

We further hypothesized that the high E2F pathway score group would demonstrate higher proliferation ability. Indeed, we found that the high E2F pathway score group demonstrated significantly higher expression of the *MKI67* gene (Figure 1B; *p* < 0.001) in both TCGA and METABRIC cohorts. High mutation rate is also a known associated factor for tumor aggressiveness, as we previously reported [54]. Hence, we investigated the possible association between the E2F pathway score and tumor burdens for neoantigens arising from mutations of type insertion and deletion (Indel) and single nucleotide variation (SNV) and copy number alteration (CNA). We found that Indel and SNV neoantigen loads were associated with a high E2F pathway score (Figure 1C; *p* = 0.006 and 0.047 respectively). Furthermore, the E2F pathway score correlated positively with CNA (Figure 1D; Spearman *r* = 0.55, *p* < 0.001). Intra-tumoral genome heterogeneity and proliferation score [57] were significantly associated with the E2F pathway score as well (Figure 1E; *p* < 0.001). Thus, the high E2F pathway score group was associated with tumor aggressiveness.

### 3.2. High E2F Pathway Score Is Associated with Worse Clinical Features

Since the E2F pathway score reflects cell proliferation ability, we hypothesized that the higher E2F pathway score would be associated with worse clinical features. Indeed, the E2F pathway score was higher in the patients with triple negative (TNBC) and HER2-positive compared to the patients with ER-positive/HER2-negative subtypes (Figure 2A; *p* < 0.001). Similarly, the E2F pathway score was higher in the more advanced American Joint Committee on Cancer (AJCC) pathological stage (Figure 2A; *p* < 0.001) and higher Nottingham histological grade (Figure 2A; *p* < 0.001). These results were consistent in both TCGA and METABRIC cohorts (Figure 2A). Furthermore, the high E2F pathway score was also associated with a higher score of each parameter of the Nottingham histological grade, among which mitotic count and nuclear pleomorphism reflected cancer cell aggressiveness (Figure 2B; *p* < 0.001, TCGA cohort). These findings suggest that the E2F pathway score represents tumor aggressiveness associated with cancer cell proliferation.

### 3.3. Immune Cell Infiltration Is Increased in Tumors with a High E2F Pathway Score

Our group and others have reported that tumor immune microenvironment (TIME) is deeply involved in cancer progression in multiple settings [58,59,60]. Cell cycle activity was recently reported to correlate with increased anti-tumor immunity in multiple cancers [61], and CDK4/6 inhibition was reported to trigger anti-cancerous immunity [62]. To this end, we hypothesized that the E2F cell cycle pathway is associated anti-cancerous immunity, and we examined the TIME of the high and low E2F pathway score group in TCGA and METABRIC cohorts to test this hypothesis. The immune cell composition of tumors was obtained from bulk tumor gene expression data with the aid of the xCell algorithm. The high E2F pathway score group demonstrated significantly higher fractions of not only pro-cancerous regulatory T cells (Tregs), helper T cell (Th2), but also anti-cancerous CD4 memory T cell, helper T cell (Th1), and M1 macrophage as well as B cells compared to the low-score group in the TCGA cohort (Figure 3A; *p* < 0.001). Similar trends were observed in the METABRIC cohort (Figure 3A). Next, we investigated the relationship between immunity and the E2F pathway score using some score derived from other methods in the TCGA cohort [57]. The E2F pathway score demonstrated a positive correlation with interferon (*INF*)–gamma response score as well as tumor infiltrating lymphocytes (TIL) regional fraction, and negative correlation with transforming growth factor (*TGF*)-beta response score (Figure 3B; *p* < 0.001). Thus, it is speculated that tumors with a high E2F pathway score have immune activation, and both pro- and anti-cancerous immune cells are infiltrated in TIME compared to the low score group.

### 3.4. Metastatic Tumors Have a Higher E2F Pathway Score than Their Primary Tumors

It is well-known that metastatic tumors have more aggressive biology such as enhanced cell proliferation compared to the primary tumors [63]. Thus, we hypothesized that metastatic tumors would have a higher E2F pathway score than the primary tumors. We previously reported that a 4-gene score derived from the tumor expression of *DOK4*, *HCCS*, *PGF*, and *SHCBP1* genes is associated with tumor aggressiveness and can be considered as a potential predictive biomarker for breast cancer [32]. First, we compared the 4-gene score between high and low E2F pathway score groups. We found that the high E2F pathway score group has a significantly higher 4-gene score (Figure 4A, *p* < 0.001) in both TCGA and METABRIC cohorts.

We compared the E2F pathway score between the primary and metastatic tumors by paired analysis (tumors from the same patient) using data of the GSE110590 cohort. Consistent with Figure 2A, the basal PAM50 breast cancer subtype demonstrated overall a higher E2F score compared with luminal or normal subtypes of primary tumors (Figure 4B). Metastatic tumors generally demonstrated a significantly higher E2F pathway score compared to the primary tumors from which they arose. However, there were a few patients who demonstrated a lower E2F pathway score in the metastatic sites: one metastatic lesion (adrenal metastasis) in the luminal type, two metastatic lesions (pancreatic metastases) in the normal type, and eight metastatic lesions (liver, lung, brain, and lymph node metastases, although seven out of eight lesions were from one patient) in the basal type. 

Given the strong association between the score and tumor aggressiveness, we hypothesized that the metastatic tumors with the high E2F score are also associated with worse prognosis. In order to test this hypothesis, we used the GSE124647 cohort, for which progression-free survival (PFS) and gene expression data are available. Metastases with a high E2F score were associated with significantly worse PFS in the whole cohort, as well as patient sub-groups with only local recurrence and with liver metastasis (Figure 4C). These findings indicated that the E2F pathway score reflects the aggressiveness of metastatic tumors as well.

### 3.5. Pre-Treatment Tumor E2F Pathway Score Is Predictive of Response to Neoadjuvant Chemotherapy (NAC) but Is Not Associated with Improved Survival 

Since the E2F pathway score reflects tumor aggressiveness, and highly proliferative cells respond to chemotherapy better, we hypothesized that the score would also reflect response to treatment, similar to tumor markers. The GSE28844 cohort was used since it is an NAC cohort for which transcriptomic data as well as treatment response data (bad, mid, good) are also available before and during NAC. We found that the pre-NAC tumor E2F pathway scores were significantly lower in the patients who had a good response to NAC as per Miller Payne criteria compared to those with mid or bad response [35] (Figure 5A; *p* < 0.001). The scores were not different between those who had mid and bad response. Additionally, we also found that the tumor E2F score decreased for patients after successful endocrine therapy with anastrozole and fulvestrant in combination with gefitinib in the GSE33658 cohort (Appendix A; *p* < 0.001). Thus, the tumor E2F pathway score was associated with treatment response among the patients who demonstrated good response to either NAC or endocrine therapy. It will be interesting to see whether there is any difference in response to aromatase inhibitor or fulvestrant by E2F score; however, that analysis was not possible due to a lack of access to data regarding which patients received what agents.

ER-positive breast cancers do not respond well to NAC compared to TNBC. Therefore, the identification of novel biomarkers to select responders to NAC among ER-positive breast cancer will increase the rate of breast-conserving surgery and avoid ineffective NAC and their side effects. We hypothesized that a high pre-treatment tumor E2F pathway score is associated with a better response to NAC. Indeed, we found that pathological complete response (pCR) rates were significantly higher in the high E2F pathway score group in the GSE25066 cohort that underwent taxane and anthracycline with endocrine therapy (Figure 5B; *n* = 467, *p* < 0.001,). Although there were similar trends, statistical significance was not achieved in two other smaller cohorts (GSE 32646 that underwent 5-fluorouracil/epirubicin/cyclophosphamide and paclitaxel (*n* = 81) and GSE 20194 that underwent paclitaxel, 5-fluorouracil, cyclophosphamide, and doxorubicin (*n* = 197). Interestingly, similar trends were observed in TNBCs, but less prominent than the ER-positive subtype. We further investigated the association between the pre-treatment E2F pathway score and disease-free survival (DFS) after NAC using the GSE25066 cohort, because pCR after NAC is generally considered a surrogate marker for better survival. Contrary to our hypothesis, the E2F pathway score demonstrated an association with DFS in either the ER-positive/HER2-negative or TNBC subtype (Figure 5C). Furthermore, although the score significantly decreased with successful endocrine therapy in the GSE33658 cohort (Appendix A) similar to NAC, a high E2F pathway score was not associated with pCR rate. In addition, the high scoring group was significantly associated with worse survival in the GSE9195 endocrine therapy cohort (Appendix A). These findings implicated that the E2F pathway score can be a predictive biomarker of NAC, but not neoadjuvant endocrine therapy in the patients with ER-positive/HER2-negative breast cancer. Furthermore, the score was not associated with survival after NAC or endocrine therapy. 

### 3.6. Patients with a High E2F Pathway Score Have Increased CDK-Related Genes and Expression of T Cell Exhaustion Markers

As shown in Figure 5, our results demonstrated that ER-positive/HER2-negative breast cancers with a high E2F pathway score are aggressive and highly responsive to NAC without a clear association with the survival. Thus, we investigated the possible hints that lead to a future therapeutic approach to improve survival in this population. We first examined the expression of the genes related to the CDK pathway, as E2F is a critical part of the CDK pathway and CDK inhibitors (CDKIs) are currently in clinical use for the patients with advanced breast cancers. 

The tumors with a high E2F pathway score were significantly associated with the increased expression of the analyzed CDK pathway-related genes, including CCNE1 (G1/S-specific cyclin-E1), CDKN2A, CDKN2D, CDK2, CDK4, and CDK6 (Figure 6A). This result was strikingly consistent in the TCGA and the METABRIC cohorts. Since we do not have access to a CDK inhibitor-treated breast cancer cohort containing gene expression and treatment response data, we examined CDK inhibitor drug sensitivity data in the Cancer Cell Line Encyclopedia [55] for five ER-positive/HER2-negative human breast cancer cell lines. The E2F pathway score demonstrated strong correlations with fold change and area under curve values in drug sensitivity assays for palbociclib (Figure 6B, Spearman *r* = 0.90 and 0.45, with *p* = 0.04 and 0.45, respectively). The E2F pathway score also showed a similar tendency with AUC values for ribociclib, but the score had an opposite trend in case of abemaciclib (Appendix A, Spearman *r* = 0.70 and −0.45, with *p* = 0.19 and 0.45, respectively). 

As the tumors with a high E2F pathway score were significantly associated with an increased infiltration of immune cells (Figure 3), the expression of immune checkpoint molecules was investigated. The high E2F pathway score group was associated with significantly elevated expression of immune checkpoint molecules, including programmed cell death 1 (PD-1), programmed death ligand 1(PD-L1), PD-L2, cytotoxic T-lymphocyte-associated protein 4 (CTLA4), indoleamine dioxygenase 1 (IDO1), lymphocyte activation gene 3 (LAG3), and tyrosine-based inhibitory motif domain (TIGIT) (Figure 6C). These results were validated with the METABRIC cohort. These findings suggested that the score might be a potential predictive biomarker for the treatment with CDK or immune checkpoint inhibitors.

## 4. Discussion

In the present study, we examined the association between the activity of the E2F pathway, measured from tumor gene expression as the GSVA score for the Hallmark E2F targets gene set, with cancer aggressiveness, metastasis, and treatment response, using data from multiple cohorts of breast cancer patients.

Tumors with high E2F pathway activity also had enriched expression of cell proliferation-related gene sets such as G2M signaling, MYC targets v1 and v2, and MITOTIC spindle. Given its association with underlying cell proliferation ability, the tumors with a high E2F pathway score were associated with aggressive clinical characteristics, such as TNBC, higher AJCC pathological stage, higher Nottingham histological grade, elevated *MKI-67* gene expression, and neoantigen loads. Furthermore, the E2F pathway had a positive correlation with a 4-gene score that we have associated with highly aggressive cancer [32]. A high E2F pathway score was associated with greater multiple immune cells infiltration by both pro- and anti-cancerous immune cell as well as B cells in the TIME. Additionally, the E2F pathway score was significantly decreased with good response to chemotherapy and demonstrated a higher pCR rate after neoadjuvant therapies in the high E2F pathway score group. Hence, it can be a potential predictive biomarker for neoadjuvant therapies in ER-positive/HER2-negative breast cancer, although the association between the score and survival benefit was unclear. Lastly, the E2F pathway score was significantly associated with the expression of CDK pathway-related genes and immune checkpoint molecules and demonstrated a positive correlation with CDKIs response.

Dysregulation of the cell cycle results in disorganized and unlimited cell growth, which is one of the hallmarks of cancer. Over-activation of the CDK4/6-cyclin D-Rb-E2F pathway has been observed in many cancers, including that of breast [64,65,66,67,68]. Rb is impaired by mutation or deletion and relieves the Rb-mediated suppression of the family of E2F transcription factors, resulting in accelerated tumor growth [66,69]. Liu and colleagues reported that the gene expression of each E2F transcription factors was associated with clinical outcomes in breast cancer [70]. Furthermore, E2F transcription factors, from E2F1 to E2F8, have been reported as possible biomarkers for breast cancer [71,72,73], although their expression patterns and prognostic significance have been inconsistent in previous studies [73,74,75,76,77,78,79]. Depending on the context, E2Fs were found to exert opposite functions as oncogenes or tumor suppressors during carcinogenesis [80]. As a result of multiple pathways and multiple E2F transcription factors involving E2F signaling, it is not practical to use only one pathway or one expression to assess the whole signaling. To overcome this issue, we defined the E2F pathway score, which summarizes the activity of the multiple genes of the E2F pathway. As demonstrated in Figure 1, the score represented underlying cell proliferation ability and was associated with other cell cycle gene sets as well. In comparison between the primary tumor and the metastatic tumor from the same patient, many metastatic tumors had higher levels of the E2F pathway score compared to the primary in PAM50 luminal or normal-type breast cancer. Metastatic tumors with the opposite results were found only at sites that are rare for metastatic lesions such as the adrenal gland or pancreas, although the result was analyzed from only one cohort.

We have recently reported that the G2M checkpoint pathway score using GSVA was associated with metastasis and poor survival in ER-positive/HER2-negative breast cancer [21]. Both G2M checkpoint and E2F target genes are involved in the cell cycle; however, each of them works on distinctively different phases of the cell cycle. The G2M checkpoint occurs between the G2 and M phases, whereas E2F plays a major role during the G1/S transition. The G2M gene set was defined as “genes involved in G2/M checkpoint”, the E2F gene set was defined as “gene encoding cell cycle-related targets for the E2F transcription factor” by MSigDB, and overlapping genes between these two pathways are less than 40%, as shown in Appendix A. In the current study, we focused on the association of the E2F pathway score with response to therapy, including neoadjuvant therapy, anti-CDK4/6, and immune checkpoint inhibition in patients with ER-positive breast cancer, which have been attracting attention in recent years.

Neoadjuvant chemotherapy (NAC) is currently a part of standard of care for patients with large primary breast tumors. However, it is well-known that the majority of patients with an ER-positive subtype—who account for more than 70% of all breast cancer patients—do not respond well to NAC, as TNBC likely due to lower cell proliferation. To this end, a predictive biomarker that accurately selects responders to NAC among ER-positive breast cancer is expected to maximize treatment benefit, reduce toxicities and financial costs, and improve patients’ quality of life. Genomic signature profiling, such as Oncotype DX and MammaPrint, has been utilized in the clinical practice to predict future recurrence risk and the benefit of adjuvant chemotherapy [81,82]. In the present study, the E2F pathway score correlated with pCR after NAC and endocrine therapy in ER-positive/HER2-negative breast cancer. To this end, the E2F pathway score will have the clinical utility that does not overlap with existing genomic signature profiling. Almost all patients in the high E2F pathway score group achieved pCR in the other three cohorts as well, although they were not statically significant, which was most likely due to the sample sizes being too small.

It was a major disappointment that the E2F pathway score did not correlate with DFS, despite a significant association with pCR after NAC and endocrine therapy in ER-positive/HER2-negative breast cancer, because pCR after NAC is considered a surrogate for better prognosis. Indeed, the number of pCR after NAC was too small in ER-positive breast cancer patients. With that said, we cannot help but speculate that cytotoxic chemotherapy or endocrine therapy was effective enough to achieve pCR in the neoadjuvant setting, but E2F pathway high-score tumors are biologically too aggressive to be controlled after they recurred or metastasized, given our results that the E2F pathway score is associated with aggressive cancer biology and the worse survival of metastatic tumors.

Recently, several clinical trials have demonstrated that the addition of CDK4/6 selective inhibitors to endocrine therapy has improved progression-free survival as well as overall survival in patients with advanced breast cancer [83,84,85]. Since CDK is considered a key molecule for several cell cycle transitions, targeting this pathway has been studied in multiple cancer types in the last decade [86,87,88,89,90,91]. However, it also remains challenging to accurately identify patients who respond to the targeted therapies. It has been reported that several genes, such as *AKT*, *CDKN2A*, *CDKN1B*, and *RAS* might serve as important potential biomarkers for response to CDK inhibitors [92,93,94,95,96,97,98], although their clinical application have not been established yet. Furthermore, immune checkpoint inhibitors, such as anti PD-1/PD-L1 antibodies, have also revolutionized the cancer treatment in multiple types of cancer, including breast cancer [99]. Although atezolizumab, an anti PD-L1 antibody, combined with nab-paclitaxel was recently approved for patients with TNBC by the US Food and Drug Administration after the IMpassion130 trial [100,101]; the overall response rate to immune checkpoint inhibitors in advanced breast cancer is generally low, around 15%, as demonstrated for other cancer types [102]. In this study, we found that the high E2F pathway score was significantly associated with the expression of CDK pathway-related genes and major immune checkpoint molecules. Furthermore, the E2F pathway score demonstrated a strong correlation with the cytotoxic effect of CDK inhibitors in the breast cancer cell lines. These findings suggest that the E2F pathway score, besides being a biomarker for response to neoadjuvant treatment, may also have a value in predicting response to CDK as well as immune checkpoint inhibitors among patients with breast cancer.

There are a few limitations to conclusions that can be drawn from our study. First, although the TCGA and METABRIC are very powerful publicly available cohorts, our study is a retrospective study. A prospective study will be needed to establish the E2F pathway score as a predictive biomarker in breast cancer management. Secondly, given the significant association between the E2F pathway score and the expression of immune checkpoint molecule CDK pathway-related gene expressions, as well as the impact of CDK inhibitors in human cell lines, further experiments and prospective studies are warranted to examine the utility of E2F pathway scoring as a biomarker of patient selection for immune checkpoint and CDK inhibitors.

In conclusion, we demonstrate that the E2F pathway score can be of value for identifying respondents to NAC among patients who have ER-positive breast cancer. Our findings also suggest a possible future use of the E2F score as a predictive biomarker for response to immune checkpoint and CDK inhibitor therapy.

## 5. Conclusions

Our findings indicated that the E2F pathway score may be a potential predictive marker of response to NAC and a predictor of CDK inhibitor and immune checkpoint therapy in patients with ER-positive breast cancer.

## Figures and Tables

**Figure 1 cells-09-01643-f001:**
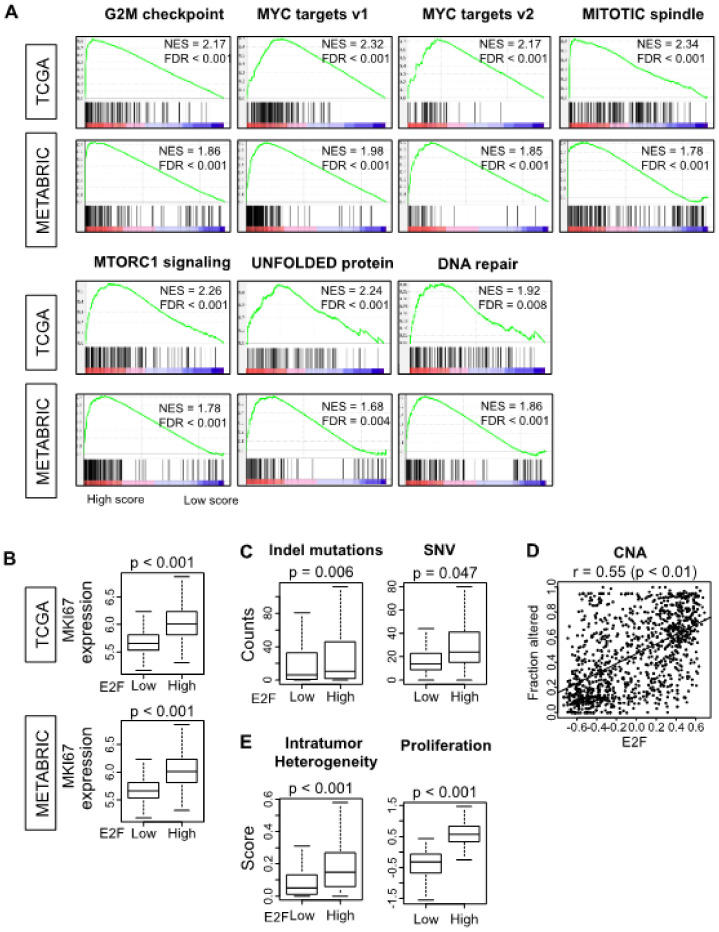
Molecular features of tumors with high and low E2F pathway scores. Within each cohort, the median value of the score is used to divide cohorts into high and low groups. (**A**) Enrichment plots along with normalized enrichment score (*NES*) and false discovery rate (*FDR*) are shown for the seven Hallmark gene sets for which enriched expression is seen in the high compared to low group in both the The Cancer Genome Atlas (TCGA) and Molecular Taxonomy of Breast Cancer International Consortium (METABRIC) cohorts. (**B**) Tumor *MKI-67* gene expression (log_2_ transcripts per million) of the two groups in the two cohorts. (**C**) Tumor genome changes of types insertion/deletion (*Indel*) and single nucleotide variation (*SNV*) in the two groups of the TCGA cohort. (**D**) Association of tumor E2F score with genome copy number alteration (*CNA*) is shown for the TCGA cohort with a scatterplot and Spearman correlation statistics. (**E**) Intratumor heterogeneity and proliferation score in the two groups. All boxplots are of Tukey type. For panels B, C, and E, *p* values in group comparison with ANOVA test are also shown.

**Figure 2 cells-09-01643-f002:**
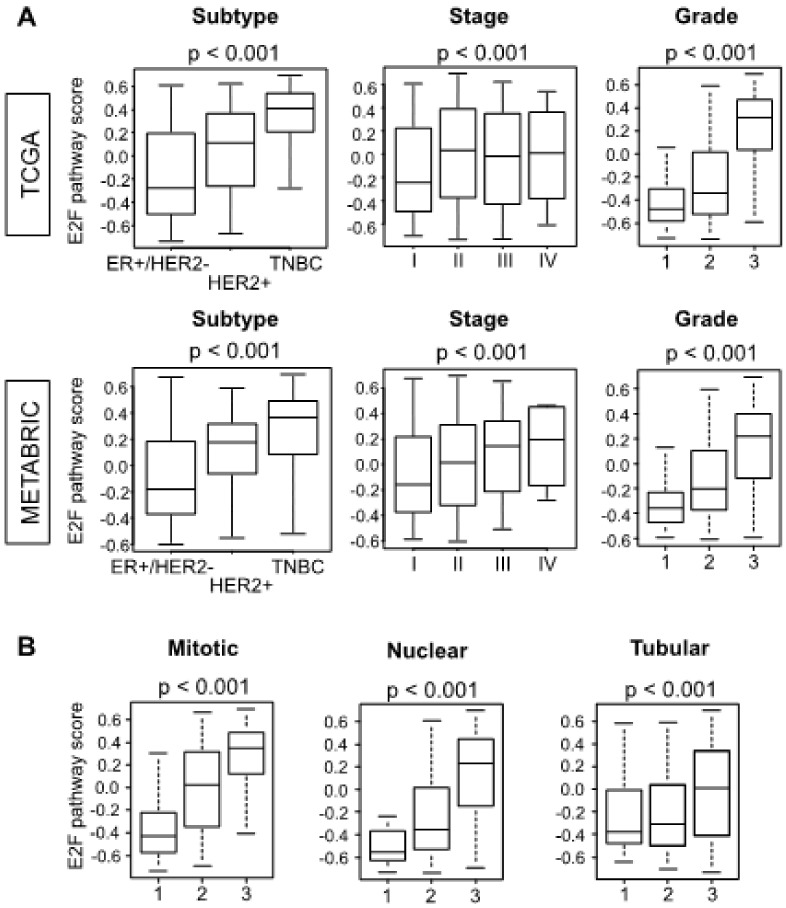
Tumors with a high E2F pathway score are more aggressive. (**A**) Boxplots of the E2F scores of tumors of different subtype, American Joint Committee on Cancer pathological stage, and Nottingham histological grade are shown for the TCGA and METABRIC cohorts. (**B**) E2F scores among TCGA tumors of different mitotic, nuclear, and tubular Nottingham sub-scores. All boxplots are of Tukey type, and depicted *p* values were calculated by ANOVA test.

**Figure 3 cells-09-01643-f003:**
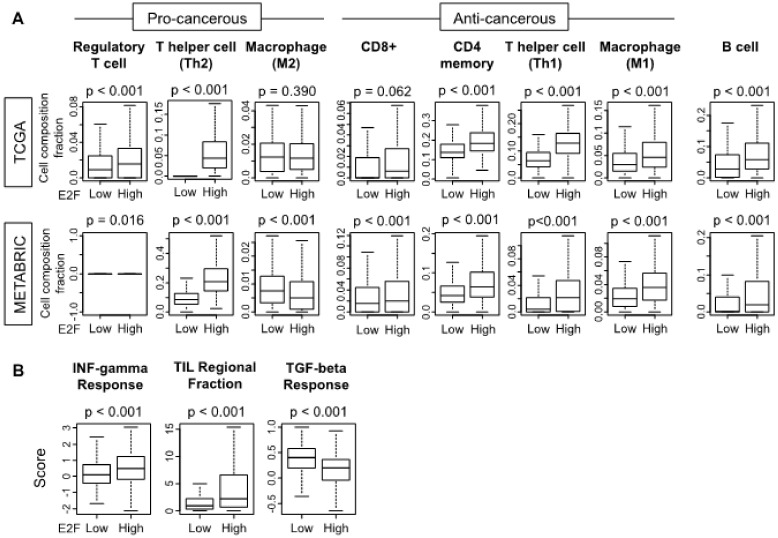
Infiltrating immune cells of tumors with high and low E2F pathway scores. (**A**) Immune cell fractions in the two tumor groups for B cells, CD8, CD4 memory, type 1 or 2 helper (*Th1/2*), and regulatory T cells, and M1 and M2 macrophages are shown with boxplots for the TCGA and METABRIC cohorts. Immune composition was derived from tumor gene expression with the xCell method. (**B**) Interferon (*INF*)–gamma response score, tumor infiltrating lymphocyte (*TIL*) regional fraction, and transforming growth factor (*TGF*)-beta response score in the two groups. Within-cohort median value of the E2F score is used to divide cohorts into high and low groups. Tukey-type boxplots demonstrate median and inter-quartile level values and depicted *p* values were calculated by the ANOVA test.

**Figure 4 cells-09-01643-f004:**
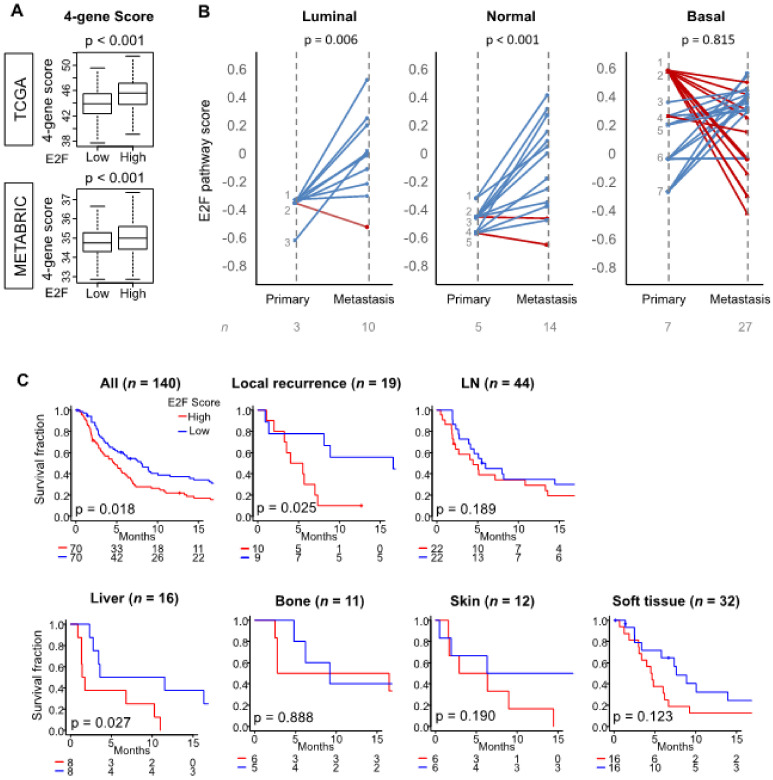
Association of tumor E2F pathway score with metastasis. (**A**) The 4-gene score values of tumors of high and low E2F scores of the TCGA and METABRIC cohorts. Tukey-type boxplots demonstrate median and inter-quartile level values, and *p* values were calculated by the ANOVA test. The 4-gene score was calculated from the tumor expression of *DOK4*, *HCCS*, *PGF*, and *SHCBP1* genes. (**B**) Matched comparison of E2F scores of primary tumors and their metastatic tumors for luminal, normal, and basal (PAM50 classification) types of breast cancer in the GSE110590 cohort. Cases with E2F scores greater for metastasis compared to the primary tumor are marked blue, and others are marked red. Group sizes are shown underneath the plots. *p* values were calculated by the paired *t* test. (**C**) Primary tumors with high and low E2F scores in the GSE124647 cohort of patients with distant metastasis or local recurrence (*n* = 140) are compared for progression-free survival within the whole cohort as well as sub-groups of patients with only local recurrence or distant metastasis to specific sites. Kaplan–Meier survival curves are compared with logrank test. The within-cohort median value of the E2F score was used to divide patients into high and low groups.

**Figure 5 cells-09-01643-f005:**
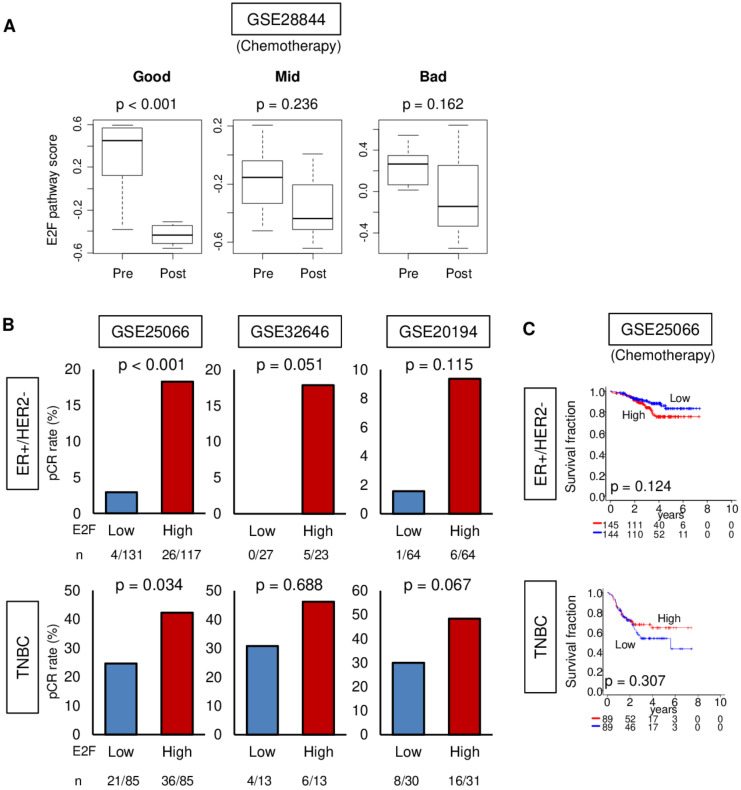
Relationships between response to neoadjuvant chemotherapy (*NAC*) and tumor E2F pathway score. (**A**) Boxplots depict tumor E2F scores before (*pre*) and after (*post*) treatment for patients of the GSE28844 (regimens: anthracycline and taxane) cohort who had good, middle (*Mid*), or bad response to treatment as Miller–Payne criteria. Boxplots are of Tukey type, and *p* values were calculated by ANOVA test. (**B**) Patients with high or low pre-treatment tumor E2F scores are compared for the achievement of pathological complete response (*pCR*) to treatment in estrogen receptor-positive/human epidermal growth factor receptor 2-negative (*ER+/HER2−*) or triple negative breast cancer (*TNBC*) sub-groups in the GSE25066 (*n* = 467, regimens: taxane and anthracycline), GSE32646 (*n* = 81, regimens: 5-fluorouracil/epirubicin/cyclophosphamide and paclitaxel), and GSE20194 (*n* = 197, regimens: paclitaxel, 5-fluorouracil, cyclophosphamide, and doxorubicin) cohorts. Numbers of patients who were treated and those who achieved pCR are noted below the plots. Rates of *pCR* were compared with Fisher’s exact test. (**C**) Patients of the GSE25066 cohort with high or low pre-treatment tumor E2F scores are compared for disease-free survival in ER+/HER2- and TNBC sub-groups. Kaplan–Meier survival curves are compared with the logrank test. The within-cohort median value of the E2F score is used to classify patients into high and low groups.

**Figure 6 cells-09-01643-f006:**
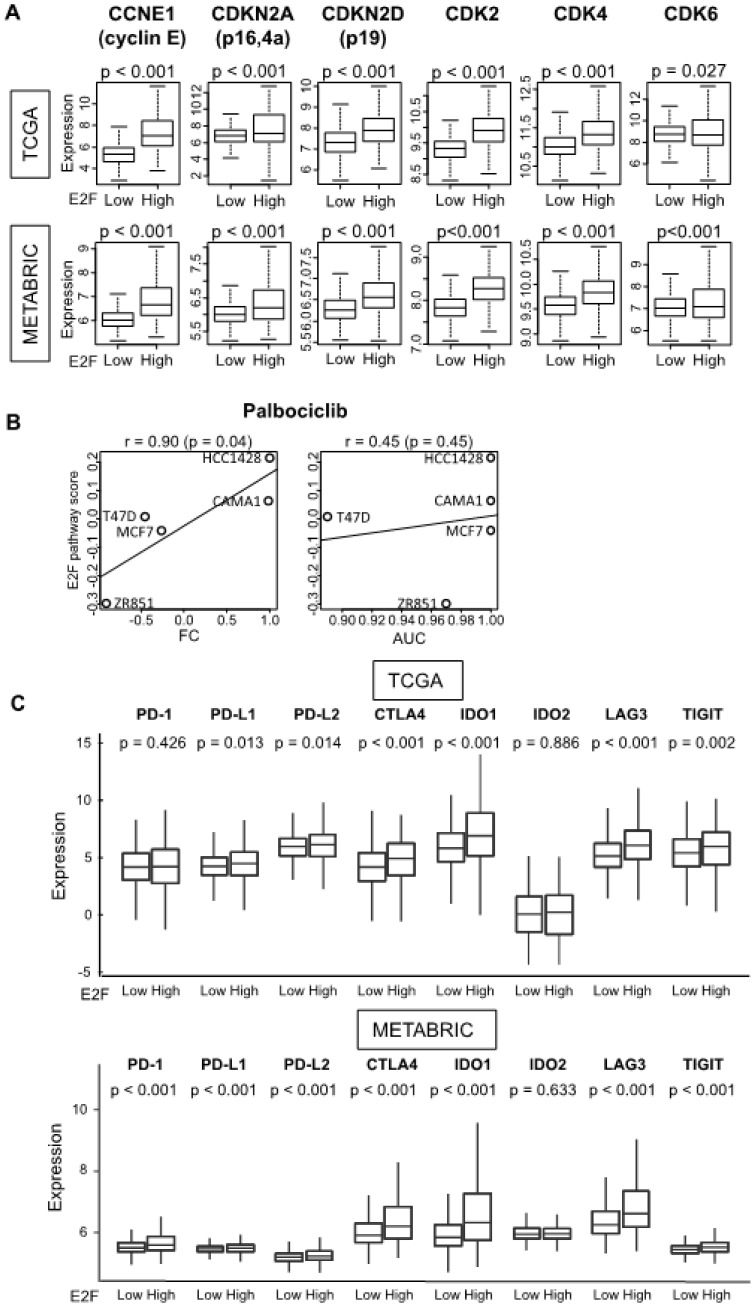
Relationships between targeted therapy and the E2F pathway score. (**A**) Boxplots depict tumor gene expression (log_2_ transcripts per million) of cyclin-dependent kinase (*CDK*) pathway-related genes of tumors with low and high E2F scores among the TCGA and METABRIC cohorts. (**B**) Association in human breast cancer cell lines between E2F score and sensitivity to CDK inhibitors. Sensitivity to palbociclib measured in fold change (*FC*) or area under curve (*AUC*) units is plotted against the E2F score of five cell lines (GSE36139 study). Spearman correlation statistics are depicted. (**C**) Tumor gene expression (log_2_ transcripts per million) of T cell exhaustion markers in the high and low E2F score groups of the TCGA and METABRIC cohorts. The median value of the E2F score is used to classify tumors into low and high groups. Tukey-type boxplots demonstrate median and inter-quartile level values, and *p* values were calculated by the ANOVA test. *CCNE1*, cyclin E1; *CDKN2A*, CDK inhibitor 2A; *CDKN2D*, CDK inhibitor 2D; *CTLA4*, cytotoxic T-lymphocyte-associated protein 4; *IDO1/2*, indoleamine dioxygenase 1/2; *LAG3*, lymphocyte activation gene 3; *PD-1*, programmed death-1; *PD-L1*, programmed death ligand 1; *TIGIT*, tyrosine-based inhibitory motif domain.

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
