# Peer review of "The E2F Pathway Score as a Predictive Biomarker of Response to Neoadjuvant Therapy in ER+/HER2− Breast Cancer"

_cells, 2020, doi:10.3390/cells9071643_

Round 1

Reviewer 1 Report

Neoadjuvant therapy consists of tamoxifen and aromatase inhbitors administration in ER positive BC tumours. What protocols did you use for neoadjuvant treatment and was there any difference in response to aromatase inhibitors and tamoxifen ??

Reviewer 2 Report

The authors investigated the E2F pathway score can be of value for identifying respondents to NAC among patients with ER-positive breast cancer. They demonstrated the possibility of the E2F score as a predictive biomarker for response to immune checkpoint and CDK inhibitor therapy. This study is well-designed by using several databases and the conclusion is supported by data reasonably.

My points are detailed below, and I hope that the authors will find them useful.

-Figure 5B. n values are hard to see because the color of these characters is dilute.

-Page 12-13 (Line 336-348). The authors used many ; (semicolon). Are they correct?

Reviewer 3 Report

In this manuscript, the authors demonstrated that the E2F pathway score can be of value for identifying respondents to NAC among patients who have ER-positive breast cancer. Several minor points should be addressed.

Minor points:
1) Why did the author study the tumor immune microenvironment of the high and low E2F pathway score groups in TCGA and METABRIC cohorts? Is there any reason encouraging the authors to study the relationship between E2F pathway with immune system? The authors may like to explain in more detail.

2) In section 2.4, the authors hypothesized that metastatic tumors would have a higher E2F pathway score than the primary tumors. What is the relationship between E2F pathway and metastatic?

3) In section 2.5, the authors used the data of the GSE28844 cohort. What’s the pre-treatment in this study? The authors may like to provide the treatment drug and explain the reasons of choosing this data for study.

4) Lastly, the authors may like to introduce the E2F pathway in the introduction. The following references may be useful:
1. Chemistry–An Asian Journal, 2018, 13(3): 275-279.2.
2. Biochemical Pharmacology, 2013, 85(3): 385-395.
3. European Journal of Medicinal Chemistry, 2018, 143: 1021-1027.
4. Autophagy, 2016, 12(6): 999-1014.
5. Nanotoxicology, 2011, 5(4): 502-516.
6. Chemical Science 8.7 (2017): 4756-4763.

Overall, I recommend publication of this manuscript after minor revision.

Reviewer 4 Report

Dr. Masanori Oshi et al. reported this study for assessing The E2F pathway score as a predictive biomarker of response to neoadjuvant therapy, especially focus analysis in ER+/HER2- breast cancer. The authors used The Cancer Genome Atlas (TCGA) and Gene Set Variant Analysis (GSVA) to verify their observation and hypothesis. Their findings indicated that the E2F pathway score could be a kind of potential predictive marker of response to neoadjuvant therapy (NAC), and predictor of CDK inhibitor and immune checkpoint therapy in patients with ER positive breast cancer. Although the results and conclusion of this paper are important, useful and rapid, the manuscript lack novelty. Because the authors used this term, methods and data reource to publish similar articles, such as “G2M Cell Cycle Pathway Score as a Prognostic Biomarker of Metastasis in Estrogen Receptor (ER)-Positive Breast Cancer—Int. J. Mol. Sci. 2020/ Apr” and “E2F cell cycle pathway score as a predictive biomarker of ER+/HER2- breast cancer response to neoadjuvant chemotherapy-- Journal of Clinical Oncology 38 (15_suppl):e12593-e12593, published online May 25, 2020.” ……combine their nearly months published articles and conclusions lead to dilute importance of this manuscript. Unless the authors provide a complete or reasonable illustration to compare what difference among these three article conclusions.

Round 2

Reviewer 4 Report

I accept the author's complement and I suggest that they should consider reducing the reference numbers.